# Circulating Metabolites as Potential Biomarkers for Neurological Disorders—Metabolites in Neurological Disorders

**DOI:** 10.3390/metabo10100389

**Published:** 2020-09-29

**Authors:** Amanda Donatti, Amanda M. Canto, Alexandre B. Godoi, Douglas C. da Rosa, Iscia Lopes-Cendes

**Affiliations:** 1Department of Medical Genetics and Genomic Medicine, School of Medical Sciences, University of Campinas (UNICAMP), Tessália Vieira de Camargo, 126 Cidade Universitária “Zeferino Vaz”, Campinas SP 13083-887, Brazil; donatti.amanda@gmail.com (A.D.); amanda.morato.canto@gmail.com (A.M.C.); bdgalexandre@gmail.com (A.B.G.); douglascescon@gmail.com (D.C.d.R.); 2Brazilian Institute of Neuroscience and Neurotechnology, Campinas SP 13083-887, Brazil

**Keywords:** amyotrophic lateral sclerosis, Alzheimer’s disease, epilepsy, multiple sclerosis, Parkinson’s disease, stroke, metabolomics, mass spectrometry, magnetic nuclear resonance

## Abstract

There are, still, limitations to predicting the occurrence and prognosis of neurological disorders. Biomarkers are molecules that can change in different conditions, a feature that makes them potential tools to improve the diagnosis of disease, establish a prognosis, and monitor treatments. Metabolites can be used as biomarkers, and are small molecules derived from the metabolic process found in different biological media, such as tissue samples, cells, or biofluids. They can be identified using various strategies, targeted or untargeted experiments, and by different techniques, such as high-performance liquid chromatography, mass spectrometry, or nuclear magnetic resonance. In this review, we aim to discuss the current knowledge about metabolites as biomarkers for neurological disorders. We will present recent developments that show the need and the feasibility of identifying such biomarkers in different neurological disorders, as well as discuss relevant research findings in the field of metabolomics that are helping to unravel the mechanisms underlying neurological disorders. Although several relevant results have been reported in metabolomic studies in patients with neurological diseases, there is still a long way to go for the clinical use of metabolites as potential biomarkers in these disorders, and more research in the field is needed.

## 1. Introduction

Neurological disorders are currently listed among the most frequent causes of mortality and severe physical or psychological impairment throughout the world [1,2,3]. They can be classified according to the underlying mechanisms and the effects on different neurological functions. Each of these diseases has specific endogenous or environmental risk factors, but genetic predisposition plays a relevant role in many of these neurological conditions [1,2,3]. Although research on neurological disorders is one of the most prolific areas in the biomedical field—and significant achievements have been accomplished in the past few decades—there is still a considerable limitation regarding the tools clinicians and scientists can use to predict the onset of neurological diseases. Similarly, the field lacks available predictors of disease prognosis and ways to monitor the response to the treatments. Thus, the search for new biomarkers of neurological disorders is still needed [1].

Biomarkers are measurements that reflect a response to normal or pathological processes [2]. These responses can manifest as functional, physiological, biochemical, or molecular interactions [2]. Biomarkers should be easily measured and quantifiable [3]; thus, an ideal biomarker should have high sensitivity and specificity to allow for the clear distinction between a normal and disease state, and it should help in monitoring disease progression and treatment response [3]. Among the many classes of molecular biomarkers, metabolites have the potential to be used individually (i.e., specific metabolites) or in combination to provide a metabolic signature of the disease [1].

### 1.1. Metabolites and Metabolomics

Metabolites are small molecules that represent ongoing biological processes in any type of biological samples, such as tissues, cells, and biofluids (blood, plasma, serum, urine, and cerebrospinal fluid (CSF)). Metabolites can influence cell responses locally or systemically and may provide insights into the mechanisms that underlie the disease process as well as disease progression [4,5,6,7]. Given that different molecular pathways can be activated under specific biological conditions, metabolites can also change depending on drug interactions and toxicity [8]. Interestingly, metabolites can cross the blood–brain barrier so that molecules found in circulation may well represent the biological phenomena that occur in the central nervous system (CNS) [9,10]. Circulating metabolites can be released into the extracellular environment, as free molecules, or inside extracellular vesicles, making it possible for metabolites to act systemically [11]. Due to these characteristics, and the possibility of being accessed in a noninvasive manner, metabolites have become potential biomarkers for disease (Figure 1) [8].

Metabolomics is the systematic analysis of metabolites; it has been applied to the discovery of disease biomarkers. Metabolomics can be used to investigate different biological systems, revealing genetic, environmental, and physiological responses associated with physiological and pathological states (Figure 2) [6,8,12,13]. The metabolic profile, especially from biofluids, may reflect the direct metabolic states of a given organism, due to the dynamic exchanging of metabolites with different tissues [8]. Therefore, in addition to identifying disease biomarkers, metabolomics may also be used for screening metabolic pathways that are putatively involved in diseases for which no specific mechanism is known [8,14].

To identify and quantify different metabolites in biological samples, the acquisition of metabolomics data should involve molecular techniques in association with adequate experimental designs and appropriate statistical processing [8,12]. The quantification of the metabolites differs remarkably from the techniques and strategies used. Relative and absolute quantification are complementary techniques for metabolic discovery and validation and can be attained by untarget and target strategies [15]. In addition, to achieve an efficient quantification of specific metabolites in metabolomic studies, high-throughput tools are needed [15,16].

Among the main analytical methods, mass spectrometry (MS) has been widely used in the qualitative and quantitative analysis of analytes in the context of metabolomics. This technique displays high sensitivity for identifying specific metabolites, and it is a high-throughput method, capable of generating massive amounts of data in the analysis of complex biological systems [8,17,18].

Each fragment is differentiated from the others in a spectrometer called the analyzer, where the mass/charge ratio (m/z) discriminates among them by different physical principles [8]. The choice of chromatographic techniques for coupling with a mass spectrometer is determined by analyte-specific chemical principles and can be divided into gas or liquid chromatography (GC and LC, respectively) [8,19]. However, LC-MS reduces the overlap of signals coming from coeluting molecules, as it leads to high-efficiency separation and allows for an increase in the detection limit and the quality of the generated spectra [20].

The higher sensitivity of metabolite detection in MS experiments (lower than 10μmol L^−1^) faces some problems involving the compatibility of molecules with a mode of ionization or detection, and might require previous steps of analysis, as separation by high- or ultra-performance liquid chromatography (HPLC or UPLC), with the aim to reduce sample complexity and enhance detection sensibility and metabolome coverage [21,22,23]. Additionally, those separation techniques also have their particularities, such as the need for the derivatization process that converts metabolites into volatile adducts in GC based experiments. All those aspects should be considered as they can influence the time of the experimental processing and limit the number of samples [24].

Since the 1980s, another technique that has been commonly used in metabolomic studies is nuclear magnetic resonance (NMR) spectroscopy [25]. This technique is based on the detection of responses from specific atomic nuclei, located in different magnetic environments, under the influence of an external magnetic field, and applying an electromagnetic pulse in the radiofrequency band [26]. Despite its limitations, this technique has advantages that still attract many researchers, such as easy sample preparation, the preservation of samples during analysis, and high reproducibility [8]. Recently, there have been increasing reports about the combined use of spectral information from both MS and NMR in studies with complex biological samples [27]. Indeed, an off-line and, most importantly, an online coupling of these different methods has contributed to more refined and reliable analyses [28,29].

These are the most common strategies used in metabolomics to specific measure metabolites in a sample; however, they do not give a general idea of the metabolic flux in the biological condition analyzed. For more dynamic analysis, it is possible to combine MS or NMR with stable isotope tracing to predict the metabolic networks and pathways and, consequently, indirectly evaluate the biological effect of the metabolic changes found [15,30,31,32].

With the recent implementation of single-cell and in vivo real-time studies, the need for more sensitive and dynamic ways of identifying and quantifying small quantities of metabolites has emerged. Thus, recent studies have evaluated the tissue metabolomics in vivo through stable isotope tracers to delimitate the systemic metabolite kinetics, physiological function, and mechanisms affected by genetic variants or metabolites of interest [15,33,34,35,36]. These techniques can be followed by the traditional metabolomic analysis with MS or NMR to measure metabolic profile [15].

Single-cell analysis has grown in the past few years due to its high specificity and ability to analyze different cell types separately [15]. However, cell separation and metabolite quantification are still problematic, requiring the development of new technologies, such as microfluid arrays and nanodevices for cell separation [37,38,39]. Moreover, new analytical methods have been developed for single-cell metabolomic and lipidomic quantification, such as matrix-assisted laser desorption ionization (MALDI), live single-cell mass spectrometry (LSC-MS), and secondary ionization mass spectrometry (SIMS) [40,41,42,43,44]. These have higher sensitivity and resolution to detect a small concentration of metabolites and are commonly associated with a subsequent MS analysis [15]. Furthermore, more sensitive metabolite quantification methods are also attractive to discriminate between different isomers, thus increasing accuracy in biomarker identification.

Beyond the challenge of analytical techniques, other steps have pivotal roles in metabolomics, such as metabolite identification, metabolic pathway association, and biological validation [21]. While databases like Human Metabolome DataBase (HMDB) [45], PubChem [46], MassBank [47], and the Lipidomic Gateway LIPID MAPS [48] can be accessed to identify metabolites and lipids, others like Kioto Encyclopedia of Genes and Genome (KEGG) [49,50], MetaCyc [51], the Small Molecule Pathway Database (SMPDB) [52], and MetaboLights [53] can be used to characterize the metabolic pathway [21].

### 1.2. Metabololites as Clinical Biomarkers

In the past few years, there has been a significant increase in identifying clinically relevant biomarkers, which show advantages over traditional clinical tests [54,55]. However, most of the metabolomic studies currently performed aim to investigate the basic mechanism of disease and drug function and are related to the early stages of developing new drugs [24,25]. Thus, it is fair to say that the field is lacking studies to identify and, especially, validate biomarkers for clinical use. One of the reasons may be related to the complex nature of identifying and developing such biomarkers for clinical use, which require standards and procedures that go well beyond a good scientific question [55]. Furthermore, given the dynamic nature of the physiological and pathological processes in the human body, which directly have an influence on biofluid composition, and the remarkable heterogeneity by which an organism reacts to physiological and pathological changes, it is not surprising that to actually identify reliable biomarkers, capable of accurately predicting different biological conditions, is a complex task [54]. Thus, to achieve the goal of identifying clinically relevant metabolic biomarkers one must rely not only on rigorous and standardized experiments, but also take into account several sources of unspecific variabilities, such as the patient’s diet, comorbidities, medicines used, and other conditions that may ultimately influence the metabolic profile [54,55].

Furthermore, clinical studies must carefully consider the question they want to solve and then define a good standard for cohort delimitation, sample collection and preparation, an analytical technique, and the best approach for statistical analysis [54,55,56]. In case-control studies, in which the aim is to look for differences between healthy and diseased individuals, the untargeted metabolomics approach is recommended, as it can provide a landscape of the systemic condition and allow integrative interpretation [55]. However, the approach requires a large number of subjects and a good validation strategy [54]. By contrast, targeted metabolomics should be preferred when evaluating the effect of a given drug in different biological systems, tissue, or time-points [54,55].

Another critical issue to consider when evaluating clinical use biomarkers is the tissue or biofluid to be analyzed [55,56]. Tissue metabolomics can provide a precise understanding of the metabolic profile of a given tissue; however, it requires an invasive procedure to obtain the appropriate samples to be studied [55]. Thus, biofluids may be used as surrogates by reflecting the metabolic status of the primary tissue involved in the disease. In neurological disorders, the access to tissue is not only always invasive, but frequently impossible, making the cerebrospinal fluid (CSF), blood (serum or plasma) and urine, alternative biofluids that can be used for metabolomic studies [57,58]. Although, the CSF is considered the most representative biofluid to study neurological disorders. It also requires invasive procedures, making blood samples (as serum or plasma) the most common biofluid used metabolomic analysis in neurological diseases [55,58]. Although the collection is the least invasive procedure, with easy storage [59], it requires complex analytical procedures, as it must be normalized for creatinine levels or urine osmolality [55]. More recently, saliva has become an interesting biofluid due to ease of access, with low cost, speed of processing, and similarity to blood compounds [55]. Studies with saliva are especially relevant in accessing large cohorts and challenging patients, such as children and the elderly population [55].

This review aims to present recent developments in research that have shown the feasibility of using metabolites as biomarkers for neurological disorders, as well as to discuss specific applications that may enter clinical practice in the near future. In addition, we will present the most relevant metabolites identified as potential biomarkers for the following neurological diseases, presented in alphabetical order: Alzheimer’s disease (AD), amyotrophic lateral sclerosis (ALS), epilepsy, multiple sclerosis (MuS), Parkinson’s disease (PD), and stroke. These are listed as amongst the most prevalent neurological disorders [1,2,3]. A list of the metabolic findings for these diseases is also presented in Table 1.

## 2. Metabolites in Specific Neurological Diseases

### 2.1. Alzheimer’s Disease

AD is the most prevalent neurodegenerative disorder in the elderly population. It has been recognized as one of the priorities in global public health by the World Health Organization (WHO) [60,61] since it accounts for 60–70% of all patients with dementia throughout the world [60,61,62].

Even though age is the strongest risk factor for dementia, an unhealthy lifestyle may contribute to the development and the rate of progression of the disease [61]. Mild cognitive impairment (MCI) is a frequent condition in individuals over 65 years old; it has a strong relationship to dementia, but with a better prognosis [63]. However, there is still no optimal way to predict which individuals with MCI will indeed develop dementia [64]. This deficiency is partly because, as with many neurodegenerative disorders, the first symptoms of AD are mild and may start several years before the full establishment of the disease [60]. Currently, the presumptive diagnosis of AD is performed using a combination of clinical findings and laboratory tests. However, it still relies heavily on observations of the progression of symptoms and signs over a long period, an approach that can delay the diagnosis significantly [60]. Despite the knowledge of the high impact of extracellular β-amyloid deposits in brain tissue, the definitive confirmation of AD is still possible only after evaluating the post-mortem tissue and its physiological alterations [65,66]. However, assessing the presence of β-amyloid deposits in brain tissue and CSF is still a good way to seek information about the disease stages and prognosis [65].

Both procedures are invasive, and the findings are usually conclusive only several years after the establishment of dementia [67,68]. Due to these findings, the search for AD biomarkers initially has been concentrated mainly in the CSF and brain tissues [64]. Indeed, these studies have reported the presence of additional biomarkers related to β-amyloid metabolism, as well as impaired calcium homeostasis, lipid metabolism, mitochondrial dysfunction, altered cell signaling, synaptic transmission, oxidative stress, and inflammation [67,69]⁠.

Recently, more metabolomic research has been applied to the search for AD biomarkers, with promising results [70]; the goal is to establish prognostic indicators and better elucidate some of the biological mechanisms that underlie disease progression. This information could, in turn, result in future treatments that minimize neuronal damage. One study evaluated amines, lipids, oxidative stress compounds, and organic acids using LC-MS and GC-MS in 127 AD patients and 121 control subjects selected by the 42 amino acid β-amyloid peptide (Aβ_42_) and total tau (t-tau) levels in the CSF [71]. The authors observed that age and gender-dependent differential metabolic expression between patients and controls [71]. They identified 26 metabolites after all statistical corrections, in which triglycerides and amines predominated [71]. From the identified metabolites, 16 lipids, aminoadipic acid (2-AAA, an intermediate of the lysine pathway), tyrosine, and other amino acids were decreased in AD patients, while sphingomyelin was increased [71]. Furthermore, the authors proposed oxidative stress compounds, mainly lysophosphatidic acid (LPA) C18:2 (the most bioactive fraction of oxidized low-density lipoprotein) and amines, mainly amino acid tyrosine, as regulatory markers and strong predictors for AD, mostly in Apolipoprotein E gene *APOE*ε4-positive AD. This proposal implies they have an important role in biochemical dysregulation related to AD.

Notably, recent studies in patients with MCI have produced intriguing results; they have aimed to identify metabolites that could assist in diagnosis. Although still preliminary, this type of study is very promising because it may improve early AD diagnosis accuracy, leading to better chances of implementing disease-modifying therapies in the future [63]. In this way, a recent study proposed that it is possible to distinguish AD patients, MCI patients, and healthy individuals by the differences in metabolites in plasma to perform a targeted quantitative and qualitative controlled assay [70]. They detected lower plasma levels of acylcarnitine, phosphatidylcholine (PC), and sphingomyelin in AD and MCI patients compared with healthy individuals. In addition, AD and MCI patients showed differences in lysophospholipid concentrations, with higher levels in AD patients and lower levels in MCI patients [70]. Furthermore, the authors proposed dodecanedioylcarnitine, dodecanoylcarnitine, and one phosphatidylcholine with a long chain and no unsaturation (PCaaC26:0) as three potential metabolites that might distinguish among AD patients, MCI patients, and healthy people. This phenomenon may be associated with a dysregulated cascade of acetyl-CoA production and release, interfering with neuron integrity [70].

One study described altered amino acid metabolism in the amnestic mild cognitive impairment (aMCI) as a potential biomarker for MCI; it might allow early intervention to avoid or slow AD progression [63]. This study included 208 elderly individuals recruited from Shanghai, China (98 aMCI patients, and 110 healthy volunteers) and was carried out through LC analysis from urine samples [63]. From the 20 identified amino acids, only arginine was significantly different between MCI patients and controls. Arginine is better represented by examining the arginine: (citrulline + ornithine) ratio; thus, the authors also analyzed citrulline and ornithine. Their data demonstrated that patients with MCI had a decreased global arginine bioavailability ratio (GABR) [63].

A comparison between 29 patients diagnosed with MCI who subsequently developed AD (MCI-AD) and 29 healthy individuals indicated that lysophosphatidylethanolamine, named LysoPE (20:0/0:0)/LysoPE (0:0/20:0), choline, and soraphen A, might be potential early AD biomarkers in plasma; all were increased in the MCI-AD group [68]. Although soraphen A is a myxobacteria product, and its interaction with AD should be better studied, it is currently known that it is related to increased short-chain fatty acids (FA) and choline levels due to its capacity to inhibit acetyl-CoA carboxylase and avoid FA elongation [68]. Moreover, the data specified choline as the best bet for a promising biomarker in early AD diagnosis; it was the only confirmed metabolite based on its pure standard. Choline plays an essential role in neurotransmitter pathways as the acetylcholine precursor, as well as in some lipids that are related to brain function, such as phosphatidylcholine [68].

Despite the large number of studies applying metabolomics in AD research, the increased relevance of the disease worldwide, demands more information regarding the initial stages, so that one can expect a better prognosis for these patients. Moreover, the necessity of a less invasive diagnostic method has been urgent in the clinic, especially when the primary analysis is performed using CSF and brain tissue. Therefore, future studies using metabolomics are expected to give us information on underlying mechanisms of the disease, as much as the discovery of biomarkers that could predict and help in the diagnosis of molecular progression and therapy response for AD patients.

### 2.2. Amyotrophic Lateral Sclerosis

ALS is a devastating neurodegenerative disease. It is the most common motor neuron disorder (MND) and is characterized by the degeneration of upper and lower motor neurons [72]. It is a fatal and progressive condition in which diagnosis is predominantly performed by clinical observation and may be delayed due to phenotypic heterogeneity [72,73,74]. The disease leads to progressive generalized paresis of bulbar, limb, thoracic, and abdominal muscles. Around 90% of the cases are sporadic, but 10% are familial; they can be inherited in an autosomal dominant, recessive, or X-linked manner [12,18,72,75,76]. Among the known pathophysiological mechanisms, oxidative stress and metabolic alterations, including lipid and glucose metabolism, have been described in patients with ALS, and deficient energy homeostasis has been shown in patients harboring the gene coding for superoxide dismutase 1 (*SOD1*) mutations [73,77,78,79,80]. Glutamate toxicity, apoptosis, and protein misfolding and aggregation are also related to the pathogenesis of ALS [8,9,10]. However, there is still a considerable lack of knowledge regarding the specific mechanism that underlies motor-neuron degeneration in ALS.

In this context, metabolomics has been used in the past few decades to identify biomarkers that might improve diagnosis and prognosis and increase understanding of the disease mechanism. A study profiled 12 elevated compounds associated with the benzothiazole drug class, commonly used to treat the disorder, on plasma samples of patients with MND, including ALS [5]. Two of the most induced metabolites were not derived from the drug itself, and thus their levels might represent a metabolic signature that reflects modified metabolic processes [5]. This study also demonstrated that MND is, in general, more associated with a metabolomic downregulation pattern [5]. Additional studies have identified several metabolites in biofluids of patients with ALS; however, the most consistent finding in these studies has been identifying glutamate as an important circulating metabolite in patients with ALS [73,78,81,82]. This finding supports the hypothesis of glutamate toxicity in ALS. Hence, one study utilized H^1^NMR spectroscopy to examine the metabolite profile of serum from ALS patients and found an increase in glutamate level [81]. Moreover, they identified reduced levels of glutamine in the same patients and proposed that these results indicate an imbalance in the glutamate–glutamine conversion cycle that occurs during excitotoxicity [81]. The authors also identified increased formate in ALS patients, data that indicate higher production of acetate [81].

Furthermore, a study from 2010 used CSF samples collected from patients with ALS at the time of diagnosis to analyze the metabolic profile of 17 compounds, including amino acids, organic acids, and ketone bodies, by NMR spectroscopy [82]. They reported reduced acetate levels and increased levels of acetone, pyruvate, and ascorbate in ALS patients. Those metabolites play a role in glutamate excitotoxicity, with involvement in brain glucose metabolism [82].

When looking at CSF samples from patients with sporadic and familial ALS, the sporadic form seems to represent a more heterogeneous condition when analyzing the metabolomic profile; indeed, it could not be discriminated from controls [78]. The familial ALS group presented *SOD*1 mutations, and it had a more distinguishable metabolomic profile [78]. These results revealed reduced glutamate and glutamine levels in the CSF of patients with ALS, especially in the *SOD*1 mutation group [78]. Another important finding was a decrease in creatinine in the CSF of both groups of patients with ALS; this change might indicate abnormal CNS energy metabolism [78]. Furthermore, a comparison of the metabolome of CSF samples of patients with ALS and other neurological disorders, using ^1^HNMR spectroscopy, confirmed the presence of increased levels of glutamic acid and ascorbic acid, which had been previously reported in the ALS group. These data indicate a possible role of the impaired glucose–lactate metabolism in ALS [73].

A study of blood plasma from ALS patients and healthy controls, using MS-targeted metabolomics, identified significantly higher levels of glutamic acid and homocysteine in the plasma of these patients [17,83]. Although those molecules are potential biomarkers for ALS, further studies are necessary to confirm their mechanisms for ALS physiology [83].

More recently, a few metabolomic studies have addressed biomarkers that could indicate a response to therapy, disease prognosis, or prediction of the risk for developing ALS. One study performed a targeted metabolomic analysis by tandem mass spectroscopy (MS/MS) in plasma samples of patients receiving or not receiving olesoxime, a chemical compound with neuroprotective properties [74], and creatinine and metabolites related to lipid metabolism were identified as the most discriminant metabolites of disease progression [74]. Moreover, metabolites involved in glutamate metabolism, especially glutamine, were also altered during disease progression in patients with ALS, independent of treatment response [74]. Glutamic acid and glycine levels were correlated to disease progression in the group receiving olesoxime treatment, and the authors described a time-dependent effect [74,84].

Furthermore, a prospective case-control study aimed to evaluate risk factors for developing ALS; it identified 31 metabolites, most of which were decreased in patients with ALS [85]. The authors’ interpretation was that these could represent biomarkers for a lower risk of developing ALS over time [85]. Some of these so-called protective metabolites include lipids, such as acylcarnitines, diacylglycerols, triacylglycerols, and phosphatidylcholines [85]. Interestingly, the authors also observed increased cholesteryl ester and phosphatidylcholines in ALS [85]. However, it is important to point out that none of the metabolites reported, as increased or decreased, reached a statistically significant threshold after corrections for multiple comparisons. Thus, these results should be interpreted with caution, and there is a need for replication in an independent sample before considering their use in clinical practice.

These metabolomic studies for ALS contributed significantly to the exploration of the molecular mechanisms underlying the disease. However, the lack of proper information about exactly how the MND in ALS happens can be a limiting factor in searching for biomarkers related to the pathological condition. Moreover, considering how devastating this disease can become, further studies aiming to find new drug targets, especially using metabolomics, could be very significant for managing patients with ALS. It would also be ideal to discover a biomarker of disease progression that could facilitate early diagnosis for these patients. However, the complexity of neurological diseases makes it very complicated to find a single factor that can serve as their definitive hallmark.

### 2.3. Epilepsy

Epilepsy is a chronic neurological disorder that affects around 50 million people throughout the world. It is characterized by the occurrence of epileptic seizures generated by abnormal neuronal discharges. An inadequate response to the currently available antiseizure medication is a widespread and critical issue in epilepsy patients. Indeed, in certain circumstances, epilepsy surgery is indicated to improve seizure control [86]. These features result in a severe economic and social burden for the patients and/or the health system [87,88]. Furthermore, the choice of medication used to control seizures in patients with epilepsy is dependent on the type of epilepsy; thus, a correct diagnosis is crucial for optimizing treatment [87]. Therefore, the search for biomarkers to assist in the diagnosis and help predict the response to treatment for epilepsy patients is urgently needed; metabolomics has been one of the tools used for these purposes [89]. However, most metabolomic studies performed on different tissues and biofluids of patients with epilepsy have also sought to contribute to the investigation of the mechanisms underlying the disease—and not only the identification of biomarkers. This dual focus is due to the fact that epilepsy is a heterogeneous condition that comprises many different specific diseases with distinct mechanisms, clinical presentations, and prognoses [86].

To the best of our knowledge, Cavus et al. were among the first researchers to analyze the metabolites of extracellular glutamate in surgical tissue from the hippocampi and cortices of patients with medically refractory epilepsy [90]. The authors aimed to investigate the energetic deficiency in these tissues, and their results suggested that it could lead to a functional impairment of glutamate transporters and glutamate reuptake [90]. A subsequent report in 2010—which used ^1^HNMR- and MS-based metabonomic analysis in blood serum from patients using different antiseizure medications—did not find any distinction in the metabolic profile of responders and non-responders to different antiseizure drugs [91]. Abela and colleagues used an untargeted metabolomics approach to identify potential plasma biomarkers in patients with a rare genetic syndrome presenting with epilepsy, specifically Snyder–Robinson syndrome [92]. They found an increase in N^8^-acetylspermidine and suggested that this metabolite could be used as a biomarker for the disease [92]. Simultaneously, another study used metabolomics to distinguish three different types of epilepsy within 48 h after seizure: partial seizure, generalized seizure, and generalized seizure secondary to partial seizure [93]. These authors used a GC-MS-based approach and found a distinct metabolomic profile in the serum of patients with epilepsy. Still, they were unable to differentiate the three types of seizures [93]. The main findings were elevated lactate and glutamate levels and reduced FAs in the serum of patients with different types of epilepsy [93]. The authors also implicated 14 endogenous metabolites altered in patients with epilepsy compared to controls; the disrupted metabolic pathways led to energy deficiencies, inflammation, and neurotoxicity [93].

In 2017, another metabolomic–genomic study aimed to identify noninvasive biomarkers using electrically mapped human cortical regions from patients with medically intractable epilepsy who underwent surgery for treatment took place [94]. They performed high-resolution magic angle spinning proton magnetic resonance spectroscopy (HR-MAS^1^H MRS), complementary DNA (cDNA) microarrays, and histological analysis in brain tissue resected during surgery. The main findings identified alterations in 14 metabolites, such as choline, glycerophosphorylcholine, glutamine, glutamate, and lactate [94]. Based on their integrated system biology approach, they suggested their results revealed an altered metabolic state in the epileptic neocortex that can be used as a future biomarker of the epileptic activity [94]. Another study analyzed serum samples from drug-resistant epilepsy patients using ^1^HNMR to identify potential biomarkers for pharmacological treatment [95]. The authors identified decreases in the concentration levels of glucose, citrate, lactate, and an increase in the concentration of ketone bodies when comparing epilepsy patients with controls [95]. Based on their observations, the authors suggested that the increase in ketone bodies might represent the body’s attempt to activate spontaneous processes to optimize the energetic resources in patients with epilepsy [95]. However, they highlighted that their results are insufficient to determine the metabolic response of drug resistance and that further studies are necessary [95]. More recently, the first metabolome-wide association study of antiseizure medication used in pregnant women was performed to understand better and potentially predict the abnormal effects of these medications in the developing fetus [96]. The authors found alterations in metabolites important for neurodevelopment and maternal health, including a decrease in neurosteroids, progesterone, and 3β-androstanediol [96]. The authors also detected a reduction in 5-methyltetrahydrofolate and tetrahydrofolate levels in patients taking lamotrigine; these changes could be related to an increased risk of neurodevelopment disorders in the exposed fetus [96].

As mentioned, a significant percentage of patients with epilepsy do not respond to antiseizure drugs, and usually, it takes many years for alternative treatments to be indicated, such as tissue resection surgery. Moreover, the surgical approach is not suitable for all types of epilepsy. Therefore, studies aiming to identify biomarkers or even molecular patterns that could predict drug response in epilepsy are very relevant. One big concern when analyzing the metabolome of epilepsy patients, as seen in other neurological diseases, is the variety of drugs these patients use, which becomes an important confounding factor for biomarker research. However, recent metabolomic studies of epilepsy patients aiming to differentiate seizure types and drug response have not succeeded. Thus, homogenizing the cohort by antiseizure and other drugs, age, sex, and other variants could increase the chances of identifying significant circulant metabolites in these patients.

### 2.4. Multiple Sclerosis

MuS is a chronic neurological disease characterized by inflammatory, autoimmune, demyelinating, and neurodegenerative events [97]. It affects the CNS, and although it is typically presented as episodes of clinical deterioration followed by periods of remission (relapsing-remitting), most patients with MuS will present irreversible neurological disabilities over time [97]. The diagnosis of MuS diagnosis relies on a combination of characteristics seen via neuroimaging studies such as magnetic resonance imaging (MRI), CSF sampling, and nonspecific clinical symptoms [97]. Given that these findings are not specific to MuS—especially in the initial stages of the disease—the diagnosis is usually delayed and often confused with other neuroinflammatory disorders. Nevertheless, evidence has suggested that early diagnosis—and, consequently, early therapeutic intervention—may attenuate the disease symptoms and further progression by decreasing the harmful effects of chronic inflammation in the CNS [98].

Metabolomic studies have been gaining increased importance in the study of MuS, mainly due to the potential use of metabolic biomarkers to assist in diagnosing MS [97,99,100]. The main biofluid source for metabolomics studies in MuS is the CSF: studies have identified increased lactate, glucose, and other metabolites that are putatively involved in energy metabolism, such as choline, acetone, myoinositol, formate, and threonate [97,101,102]. However, these same studies have also found decreased citrate levels, mannose, acetate, and phenylalanine when comparing patients with MuS and healthy controls [103,104]. Recently, an MS-based metabolomics study identified differences in tryptophan, phenylalanine, and pyrimidine metabolism in the CSF from patients with MuS, progressive, and relapsing-remitting disease, in comparison with controls [105].

A few studies have also used serum and plasma NMR-based metabolomics in patients with MuS; they have mainly identified energy metabolism changes, redox, and xenobiotic metabolism associated with the disease. The identified metabolites included glucose, valine, scylloinositol, glutamine, acetate, glutamate, lactate, and lysine [106,107]. Analysis of serum samples of a large prospective cohort of patients with MuS established that most of the altered metabolites are involved with lipid and energy metabolism, more specifically increased levels of free non-esterified FAs and FA oxidation products. They also identified phospholipid metabolites [108]. More recently, a clinical trial in 2019 used a blood-based untargeted approach to search for MuS biomarkers that identify 12 candidate metabolites in patients. Pyroglutamate, laurate, N-methylmaleimide, acylcarnitine C14:1, and phosphatidylcholine were among these metabolites. The authors also found metabolites involved in biological pathways like glutathione metabolism, cellular membrane composition, FA metabolism, and oxidation [109].

An interesting recent work proposed a new method for studying lipidomics and metabolomics in tears of patients with MuS [110]. The researchers developed a new LC-MS/MS approach that identified polar lipids, acylcarnitines, free carnitine, and amino acids from the tears. They performed a pilot experiment in patients with MS and identified 30 tear lipid biomarkers, some of which had also been found in the CSF of these patients [110]. In addition, they saw a significant increase in the acylcarnitine levels, an increase in C5OH/C4DC, C10:1, and C8:1, a decrease in C12 and C14:1, and a reduction in C18:1OH levels in the tears of patients with MuS [110].

As multiple sclerosis presents symptoms that are easily confounded with other diseases, especially inflammatory ones, it would be of great interest if metabolomics could be applied to studies aiming to differentiate these diseases with overlapping symptoms, in order to facilitate the correct diagnosis and, consequently, the right treatment for these patients.

### 2.5. Parkinson’s Disease

PD is characterized by the neurodegeneration of dopaminergic neurons from various areas of the CNS, mainly in the *substantia nigra* [111,112]. PD affects about 1–3% of the global population over 60 years of age [111]. A PD diagnosis is still essentially based on the patient’s clinical history, neurological exam, and response to dopaminergic drugs [111]; this approach can lead to misdiagnosis, especially in the early stages of the disease [111]. Clinically, the most common features are muscle stiffness, hypokinesia, gait disturbances, and resting tremor. These are slowly progressive; however, the clear clinical picture starts after the loss of about 80% of striatal dopaminergic neurons or 30–50% of these neurons in the *substantia nigra*, leading to a critical decrease in dopamine levels [111,113,114,115]. There is also a progressive accumulation of intracellular inclusions, named Lewy bodies, in different brain regions, as well as in the CSF [111,112]. These aggregates contain a protein called α-synuclein [111].

Although there are rare monogenic forms of PD—which are associated with overexpression of α-synuclein or mutations in the parkin (*PRKN*) gene [111,116,117,118,119,120]—most patients with PD do not report a family history and are considered sporadic cases with unknown etiology [116,117,118,121]. Thus, identifying biomarkers that could be related to the underlying mechanisms that lead to PD is important for assisting in the diagnosis and defining specific therapies that target the different disease mechanisms. Hence, several reports have identified mitochondrial dysfunction caused by the accumulation of reactive oxygen species (ROS) in PD [112,122,123,124]. An increase follows this ROS accumulation in α-synuclein aggregation and the typical Lewy bodies [112]. The decrease in antioxidant activity in the depletion of ROS can be attributed to several factors: the environmental exposure to toxic substances, such as pesticides, and the high oxidation potential of the dopamine itself within dopaminergic neurons [112,125]. In addition, there are other possible mechanisms related to energy metabolism that seems to also contribute to the development of PD, such as reductions in levels of the electron transport chain complex I and mutations in the gene coding for phosphatase and tensin homolog PTEN-induced kinase 1 (*PINK*1), which codes for an alleged mitochondrial kinase involved in mitochondrial fission and fusion processes, mitophagy, and oxidative phosphorylation [123,126,127,128,129,130,131,132]. Thus, mitochondrial metabolism seems to be a key mechanism involved in PD [132].

Metabolomic studies in PD have been performed in blood serum from patients. In a 2015 study, Hatano et al. identified several metabolites that were significantly decreased only during the initial stages of the disease in patients with PD, such as caffeine and its derivatives and biomolecules related to purine metabolism [133]. The potential protective action of caffeine against PD may be related to its mechanism of inhibition of adenosine A2A receptors, which regulate the release of neurotransmitters such as glutamate and dopamine in the CNS [133,134,135]. Moreover, some studies have used other analytical approaches and have identified decreases in other metabolites that belong to purine metabolisms, such as uric acid and its intermediates, contributing to an increase in cellular oxidative damage and PD progression [136,137,138]. Oxidative damage has also been demonstrated in patients with PD due to the decreased bilirubin/biliverdin ratio in the blood serum of these patients [133]. Biliverdin, a product of oxygenation and reduction reactions of heme, and bilirubin, a reduced biliverdin product, are highly regulated by the amounts of ROS. So, the biliverdin concentration is indicative of the levels of these oxidant agents. Furthermore, decreased levels of antioxidant agents, bilirubin, and ergothioneine have been reported in the blood serum of patients with PD as in other neurodegenerative diseases [133].

FAs and their products derived from beta-oxidation (FAO) have also been implicated in the development of PD. FAs such as the medium-long chain, phosphatidylcholine, and lysophosphatidylcholine are increased in the urine of patients with PD, so they have been proposed as candidates for biomarkers of PD progression [111,139,140]. The increase in these FAOs has been linked to the onset of mitochondrial dysfunctions and neuronal loss. Moreover, FAOs play a role in signaling for both cell viability and apoptosis [111,141]. Furthermore, FAO products, such as myristoleoylcarnitine, hexanoylcarnitine, decanoylcarnitine, and others, are also increased in the blood serum of patients in the early stages of PD [141,142]. Other FAOs, such as furoylglycine, malonylcarnitine, and hexanoylglycine, have also been shown to be increased in the urine of patients with PD [143,144].

Another important class of metabolites that have been relevant in PD are amino acids, such as branched-chain amino acids (BCAAs), which have been increased in animal models and patients with PD [6,143,145]. This set of metabolites comprises leucine, isoleucine, and valine, and they are known to perform functions related to energy metabolism, regulation of protein synthesis, and prevention of oxidative damage [146]. Interestingly, there is evidence for a relationship between altered BCAA metabolism and the onset of mitochondrial dysfunctions through the generation of toxic metabolites of these amino acids [147].

Altogether, the findings described above indicate that the abnormalities in purine compounds, FAs, BCAAs, bilirubin, and biliverdin—found in the biofluids of patients with PD—point towards mitochondrial dysfunction and oxidative damage as key mechanisms involved in the development and or progression of PD [111,133,148]. However, it is important to note that chronic use of the drug levodopa by patients with PD may change serum BCAA levels because they are carried across the blood-brain barrier by the same carriers that allow levodopa and aromatic amino acids to pass into brain tissue [111,149]. Hence, changes in BCAA, and perhaps other compounds, may result from specific treatments.

Aromatic amino acids, represented by tyrosine, tryptophan, phenylalanine, and histidine, are substrates for synthesizing several neurotransmitters [150]. In patients with PD, elevated levels of monoamine metabolism products, such as 3-methoxytyrosine, 3-methoxytyramine sulfate, and homovanillic acid, have been found in several biofluids, including plasma, serum, and urine [133,151]. However, such patients were under treatment with levodopa and decarboxylase inhibitors, which can influence the metabolic profile biotransformation pathways of aromatic amino acids, such as tyrosine and tryptophan [145,151,152,153]. Nevertheless, tyrosine, a precursor to both dopamine and levodopa, was found at similar levels in healthy individuals and patients with PD, independent of levodopa or a levodopa equivalent dose [133]. Furthermore, a study found that a reduction in tryptophan and serotonin levels in serum samples from patients with PD could be related to anxiety and depression symptoms, which usually appear in more advanced stages of the disease [133]. By contrast, phenylalanine metabolism components, such as phenylacetate [133] and aspartyl phenylalanine benzoate [139], and histidine metabolism compounds, such as 1-methylhistidine [142], are increased in serum samples in patients with PD.

Moreover, there are known common features among different neurodegenerative disorders. Indeed, PD and ALS are linked to protein misfolding and the consequent aggregation of biomolecules [154], which is aggravated by mitochondrial dysfunction and oxidative imbalances [77]. Therefore, it is not surprising that a study identified a similar metabolic profile in CSF and blood plasma of patients with PD and ALS [6]. In the CSF, there was an increase in ⍺-hydroxybutyric acid, *β*-hydroxybutyric acid, and carnitine, and a decrease in ammonia and uracil in both groups of patients [6]. In the blood plasma, there was an increase in xanthine, carnitine, ⍺-hydroxybutyric acid, and lactate and a decrease in hippurate, adenine, hypoxanthine, and lysophosphatidylcholine (14:0) [6].

Parkinson’s disease is another neurological disease that could benefit from metabolomic studies applied to new drug target discovery. The described research has used metabolomics to exploit the pathological mechanisms involved in developing the symptoms, but biomarker studies are still scarce. Studies aiming to predict PD development would be a significant contribution to the clinic since, nowadays, the patient has already lost around 80% of their striatal dopaminergic neurons at the time of diagnosis. Therefore, metabolomics is a potential tool in investigating molecular mechanisms related to PD and can be highly applied to drug discovery targets.

### 2.6. Stroke

Stroke is one of the leading causes of mortality and disabilities around the world [155]. According to the WHO, it is prevalent in all developed and underdeveloped countries; it significantly burdens patients, families, and the public health care system [155]. A stroke occurs when there is an obstruction of blood flow to the brain, a phenomenon that affects its function and leads to cell and tissue damage and death [156,157]. The blockage of oxygen supply to the brain causes cell death, releasing toxic chemicals, leading to further tissue damage [9]. Stroke can be classified into two main subtypes: ischaemic (IS) and hemorrhagic (HS) [156]. IS is the most frequent type; it is characterized by an interruption to the blood flow due to clots and artery blocks [156]. This subtype has both an acute and a chronic phase, depending on the time after the initial symptoms; the two phases show differences in the type of molecular pathways activated and of metabolite profile [158]. Several biological phenomena occur immediately after a stroke, including cell death, inflammation, changes in energy metabolism, increased oxidative stress, decreased cell viability, and brain damage [158,159].

The main risk factors for stroke are advanced age, high blood pressure, diabetes, dyslipidemia, and smoking [157]. However, biologic risk factors influenced by genetic predisposition are also relevant; hence, even though advanced age is one of the major risk factors, stroke incidence is increasing in young individuals [160]. An increased level of an inhibitor of the nitric oxide synthase metabolite, dimethylarginine, has been positively associated with early-onset stroke [161]. Nitric oxide synthase is part of the pathogenesis of atherosclerosis [161]. Dyslipidemia is a leading cause of carotid stenosis, which puts one at a higher risk for atherosclerotic stroke. In the severe class of carotid stenosis patients, researchers have described a decrease in choline and increased homocysteine and lipid levels [162]. Choline reduction increases the effect of the homocysteine methylation pathway and choline’s antioxidant property, which exacerbates atherosclerosis risk [162]. Increased homocysteine levels have been described as associated with an increase in oxidative stress in vascular cells and an increase in platelet adhesion to vessel walls and can be considered a biomarker for stroke [163].

Furthermore, the downregulation of some metabolites, such as L-homoarginine, cadaverine, 2-oxoglutarate, nicotinamide, valine, and ubiquinone, has also been associated with an increased risk of stroke [164,165,166]. Homoarginine has been described as part of nitric oxide metabolism and can regulate blood flow and pressure, cell viability, insulin secretion, and platelet aggregation [164]. A few studies have shown that a decrease in homoarginine levels can be associated with increased stroke mortality [164,166]. In addition, the reduction of lysine catabolites has been associated with poor stroke outcomes [165].

Wang et al. [6] evaluated the serum of 40 patients a week after IS and 29 controls using MS. The authors identified 18 metabolites related to inflammation, energy deficits, oxidative stress, neurotoxicity, neuroexcitation, and brain injury mechanisms [10]. The authors indicated altered levels of aromatic amino acids, an upregulation of lactate, and a downregulation of tyrosine and tryptophan as potential biomarkers for acute IS [10]. In stroke patients, low levels of tyrosine can increase oxidative stress and inflammation, while low levels of tryptophan can reduce serotonin [10]. Decreases in tryptophan and glutamate levels have been considered potential post-stroke depression indicators [10,167]. The authors also reported that high levels of glutamate might be involved in increasing neuronal deaths and the enlargement of infarction size in those patients [10].

The expansion of cerebral injury can also be influenced by oxidative stress due to the overexpression of matrix metallopeptidase 9 (MMP-9) [166]. A targeted study developed by Kelly et al. associated elevated MMP-9 with augmented neuronal arachidonic peroxidation metabolite (F2 isoprostanes (F2Ips)) in stroke patients. These data indicate an increase in oxidative stress processes and a consequently worse stroke outcome [163,166].

Another group of metabolites, BCAA, was also identified as altered in patients with stroke. Two studies identified decreased BCAA levels in patients with acute IS compared with controls [9,168]. Thus, these metabolites might influence the bioenergetic homeostasis and citric acid cycle pathways and might be responsible for increasing stroke severity [9,10,163].

Liu et al. reported that the combination of five metabolites—the downregulation of serine, isoleucine, and PC 50/50 and the upregulation of betaine and lysoPE(18:2)—in serum samples of patients after stroke might be a predictor of a good outcome [9]. According to the authors, the downregulation of serine, isoleucine, and aspartate in acute IS might regulate the growth, repair, and maintenance of the brain. In contrast, the upregulation of LysoPE might repair the cell membrane [9].

There are different IS types, which are classified according to the etiology of the neurovascular events and the clinical and research data. This classification is called Trial of Org 10172 in Acute Stroke Treatment (TOAST) [169]. Each subtype has different etiologies, and thus specific metabolites may be present in the different TOAST sub-types. Jung et al. reported specific changes in the metabolic profile of patients with small vessel disease with an increase in lactate, pyruvate, glycolate, and formate, as well as a decrease in glutamine and methanol secretions [170]. These metabolites have a relevant function on the oxidative stress process by anaerobic glucose and folic acid metabolisms [163,170].

Inflammation is another relevant process that has been identified in different stroke subtypes. Jove et al. reported the increased levels of the phospholipid lysophosphatidylcholine (22:6) (LysoPC(22:6)) in patients with extensive artery atherosclerosis [171]. In addition to the phospholipids phosphatidylethanolamine (PE) and LysoPC, this signaling molecule has a relevant function on neuronal membrane stabilization. It might be a potential trigger of the brain inflammation processes that might influence tissue response after stroke [163]. Another metabolite that might affect the response after stroke is taurine. High levels of taurine have been reported in patients during the acute IS phase, leading to increased brain tissue damage brain [172]. However, the right taurine dose could be used as a potential stroke treatment to reduce neuronal cell loss, apoptosis, and infarct size [172].

HS affects around 10–15% of all patients with stroke; it has been described as a rupture on brain arteries, causing a severe hemorrhage in the tissue [155,156]. HS is known to have the worst prognosis when compared with IS [155,156]. High levels of glutamate and histidine reportedly impair stroke recovery, leading to worse outcomes in patients with subarachnoid hemorrhage, a type of HS [173,174]. It is believed that the excess of these metabolites causes an increase in excitatory responses and release of free radicals, creating severe oxidative stress and progressive cell membrane degradation [163,173]. Furthermore, the high lactate: pyruvate ratio has been linked to poor prognosis after a subarachnoid hemorrhage, becoming a useful biomarker for the development, outcome, and severity of the hemorrhagic event [173,174]. Lactate and pyruvate interfere with anaerobic glycolysis response and can be indicative of ischemia. Low levels of pyruvate can adversely affect the Krebs cycle and harm the production of amino acids that are important to repair brain injury [174].

The increase in free amino acids in CSF during hospitalization has also been proposed as a marker for the catabolism process. One study reported the L form of 2-hydroxyglutarate as a biomarker for hypoxia response during hemorrhagic brain events [175]. This metabolite increase reportedly predicted an adverse outcome, including leading to patients’ death [175]. By contrast, decreased levels of the L form of 2-hydroxyglutarate have been associated with low disability among patients with HS [175]. Lu et al. also reported that an increase in glycine and proline during hospitalization of patients with subarachnoid hemorrhage might induce meningeal collagen synthesis and serve as a biomarker of the disease [175].

Metabolomics studies have contributed much to the understanding of the biology underlying stroke occurrence. Despite all the efforts to distinguish the metabolome of stroke types, improvements in case-controls cohort characterization could result in more informative and reliable data. Risk factors and drug usage standardization might lead to more precise results that could be implemented in the clinic by improving the prognosis and the treatment given to the patients after the stroke. Moreover, metabolomic studies involving multiple fields on stroke recovery are setting out to determine the best treatment for each kind of stroke [147].

**Table 1 metabolites-10-00389-t001:** Metabolites discovered for each of the main neurological diseases: Alzheimer’s disease (AD), amyotrophic lateral sclerosis (ALS), Epilepsy, multiple sclerosis (MuS), Parkinson’s disease (PD), Stroke.

Condition	Biological Fluid	Type of Analysis	Metabolite	Biomarker for	Related Mechanisms	Ref.
AD	CSF	LC-MS/ GC-MS	Aminoadipic acid	AD prediction	Not clear for this condition	[69]
	CSF	LC-MS/ GC-MS	Tyrosine	AD prediction	Neurotransmitter synthesis	[69]
	CSF	LC-MS/ GC-MS	Sphingomyelin	AD prediction	Membrane Constitution	[69]
	CSF	LC-MS/ GC-MS	Lysophosphatidic acid C18:2	AD prediction	Oxidative stress	[69]
	Plasma	FIA/ MS/MS	Acylcarnitine	AD and MCI prediction	Cascade of neurodegeneration	[68]
	Plasma	FIA/ MS/MS	Phosphatidylcholine	AD and MCI prediction	Cascade of neurodegeneration	[68]
	Plasma	FIA/ MS/MS	Sphingomyelin	AD and MCI prediction	Not clear for this condition	[68]
	Plasma	FIA/ MS/MS	Lysophospholipids	Differentiate AD from MCI	Not clear for this condition	[68]
	Plasma	FIA/ MS/MS	Dodecanedioyl carnitine	Differentiate AD from MCI from healthy subjects	Not clear for this condition	[68]
	Plasma	FIA/ MS/MS	Dodecanoylcarnitine	Differentiate AD from MCI from healthy subjects	Not clear for this condition	[68]
	Plasma	FIA/ MS/MS	PCaaC26:0	Differentiate AD from MCI from healthy subjects	Not clear for this condition	[68]
	Urine	LC	Arginine	aMCI prediction	Protein homeostasis, taurine metabolism, glutathione metabolism	[63]
	Plasma	UPLC-MS/MS	Lysophosphatidyl ethanolamine	MCI-AD prediction	Membrane Constitution	[66]
	Plasma	UPLC-MS/MS	Choline	MCI-AD prediction	Neurotransmitter synthesis	[66]
	Plasma	UPLC-MS/MS	Soraphen A	MCI-AD prediction	It can interfere in the fatty acid elongation	[66]
ALS	Serum/plasma; CSF; Plasma	NMR-based/MS-target; NMR-based; FIA/LC-MS/MS/NMR-based	Glutamate	ALS prediction; Differentiation from other neurological disorders; Drug responsiveness	Glutamate excitotoxicity	[17,71,72,78,81,82]
	Serum; CSF; plasma	NMR-based; CG-MS; FIA/ LC-MS/MS	Glutamine	ALS prediction; Familial ALS prediction (SOD1 mutation); Drug responsiveness	Imbalance in glutamate–glutamine cycle	[72,74,78]
	Serum	NMR-based	Formate	ALS prediction	Increased levels may cause cell death	[78]
	CSF	NMR-based	Acetate	ALS prediction	Energy metabolism dysfunction	[5]
	CSF	NMR-based	Acetone	ALS prediction	Energy metabolism dysfunction	[5]
	CSF	NMR-based	Pyruvate	ALS prediction	Energy metabolism dysfunction	[5]
	CSF	NMR-based	Ascorbate	ALS prediction; Differentiation from other neurological disorders	Oxidative stress	[5,71]
	CSF	CG-MS	Creatinine	Familial ALS prediction - SOD1 mutation	Energy metabolism dysfunction	[74]
	Plasma	MS-target	Homocysteine	ALS prediction	Not clear for this condition	[17,81]
	Plasma	FIA/LC-MS/MS	Creatinine	Drug responsiveness	Not clear for this condition	[72]
	Plasma	FIA/LC-MS/MS/NMR-based	Glycine	Drug responsiveness	Changes in its levels can affect the activity of the NMDA receptor	[72,82]
	Plasma	LC-MS/MS	Acylcarnitines	Protective function	Not clear for this condition	[83]
	Plasma	LC-MS/MS	Diacylglicerols	Protective function	Not clear for this condition	[83]
	Plasma	LC-MS/MS	Triacylglicerols	Protective function	Not clear for this condition	[83]
	Plasma	LC-MS/MS	Phosphatidylcholine	Protective function	Not clear for this condition	[83]
Epilepsy	Plasma	LC-MS	N8-acetylspermidine	Snyder–Robinson syndrome	Alterations in its levels may cause an imbalance of excitatory and inhibitory mechanisms	[90]
	Serum; Brain tissue; Serum	CG-MS; HR-MAS¹H MRS; NMR-based	Lactate	Different types of seizures; Epileptic activity; Drug responsiveness	Energy metabolism dysfunction	[91,92,93]
	Serum; Brain tissue	CG-MS; HR-MAS¹H MRS	Glutamate	Different types of seizures; Epileptic activity	Glutamate excitotoxicity and hyperexcitability	[91,92]
	Brain tissue	HR-MAS¹H MRS	Choline	Epileptic activity	Alterations in its levels may suggest heightened cell membrane turnover in high-spiking tissue	[92]
	Brain tissue	HR-MAS¹H MRS	Glycerophosphorylcholine	Epileptic activity	Alterations in its levels may suggest heightened cell membrane turnover in high-spiking tissue	[92]
	Brain tissue	HR-MAS¹H MRS	Glutamine	Epileptic activity	Not clear for this condition	[92]
	Serum	NMR-based	Glucose	Drug responsiveness	Energy metabolism dysfunction	[93]
	Plasma	LC-HRMS	Neurosteroids	Effect of medicines in fetal development	Neurodevelop- mental functions	[93]
	Plasma	LC-HRMS	Progesterone	Effect of medicines in fetal development	Reduced levels may be related to a risk factor for miscarriage	[93]
	Plasma	LC-HRMS	3β-androstanediol	Effect of medicines in fetal development	Not clear for this condition	[93]
	Plasma	LC-HRMS	5-methyltetrahydrofolate	Effect of medicines in fetal development	AED-induced effect on folate uptake or metabolism	[93]
	Plasma	LC-HRMS	Tetrahydrofolate	Effect of medicines in fetal development	AED-induced effect on folate uptake or metabolism	[93]
MuS	CSF; Serum	NMR-based	Acetate	MuS prediction; Differentiate Neuromyelitis optica from MuS and healthy subjects	The decrease may lead to myelination dysfunction; Neurotransmitter synthesis and suggested as a marker of astrocyte metabolism	[101,104,107]
	CSF	NMR-based	N-Methyl metabolites	Demyelination process	Impairment in the choline-glycine cycle and myelin synthesis	[101]
	CSF	NMR-based	Sarcosine (N-methyl-glycine)	Demyelination process	Impairment in the choline-glycine cycle and myelin synthesis	[101]
	CSF	NMR-based	Formate	Demyelination process	Impairment in the choline-glycine cycle and myelin synthesis	[101,102]
	CSF	NMR-based	Lactate	MuS prediction	The increase was related to CSF mononuclear cells in MS patients and demyelinating areas	[102]
	CSF	NMR-based	N-acetyl aspartate (NAA)	Differentiate chronic lesions from healthy subjects	The decrease may be related to chronic demyelinating plaques	[102,103]
	CSF	NMR-based	Choline	Differentiate acute from chronic plaques and normal-appearing white matter	Increase related to active demyelinating plaques	[102]
	CSF	NMR-based	Citrate	MuS prediction	The decrease can be related to the disruption of the TCA cycle through the pyruvate pathway and the formation of myelin	[102,103,104]
	CSF	NMR-based	Threonate	MuS prediction	Not clear for this condition	[103]
	CSF	NMR-based	Myo-inositol	MuS prediction	Not clear for this condition	[103]
	CSF	NMR-based	Mannose	MuS prediction	Not clear for this condition	[103]
	CSF	NMR-based	Phenylalanine	MuS prediction	Not clear for this condition	[103]
	CSF	NMR-based	3-hydroxybutyrate	MuS prediction	Not clear for this condition	[103]
	CSF	NMR-based	2-hydroxyisovalerate	MuS prediction	Not clear for this condition	[103]
	CSF	NMR-based	2-hydroxybutyrate	MuS prediction	The increase may be related to raised lipid oxidation and oxidative stress	[104]
	CSF; Serum	NMR-based; GC-MS	Pyroglutamate	MuS prediction	The increase may be related to impairment in antioxidant pathways and leads to central nervous system dysfunction	[104,109]
	CSF	NMR-based	Acetone	MuS prediction	The increase may be related to impairment in energetic metabolism	[104]
	CSF; Serum	NMR-based	Glucose	MuS prediction	The decrease can be related to disturbed energy generation and progress of MS	[104,106]
	CSF	HRMS	Kynurenate	Differentiate SPMuS from RRMuS patients	Tryptophan metabolism	[105]
	CSF	HRMS	5-hydroxytryptophan	Differentiate SPMuS from RRMuS patients	Tryptophan metabolism	[105]
	CSF	HRMS	5-hydroxyindoleacetate	Differentiate SPMuS from RRMuS patients	Tryptophan metabolism	[105]
	CSF	HRMS	N-acetylserotonin	Differentiate SPMuS from RRMuS patients	Tryptophan metabolism	[105]
	CSF	HRMS	Uridine	Differentiate SPMuS from RRMuS patients	Pyrimidine metabolism; Significantly associated with disability, disease activity, and brain atrophy	[105]
	CSF	HRMS	Deoxyuridine	Differentiate SPMuS from RRMuS patients	Pyrimidine metabolism; Significantly associated with disability, disease activity, and brain atrophy	[105]
	CSF	HRMS	Thymine	Differentiate SPMuS from RRMuS patients	Pyrimidine metabolism; Significantly associated with disability, disease activity, and brain atrophy	[105]
	CSF	HRMS	Glutamine	Differentiate SPMuS from RRMuS patients	Pyrimidine metabolism; Significantly associated with disability, disease activity, and brain atrophy	[105]
	Serum	NMR-based	Selenium	MuS prediction	The decrease may be related to oxidative stress	[106]
	Serum	NMR-based	Valine	MuS prediction	The decrease may be related to myelination dysfunction of the neurons	[106]
	Serum	NMR-based	Scyllo-inositol	Differentiate MuS from Neuromyelitis optica and healthy subjects	May be related to diffuse glial proliferation, demyelination, and neuronal damages	[107]
	Serum	UHPLC-MS	Sphingomyelin	MuS prognosis	One of the main lipid class in myelin; influence the immune response	[108]
	Serum	UHPLC-MS	Lysophosphatidyl ethanolamine	MuS prognosis	Modulates the immune response	[108]
	Serum	UHPLC-MS	Hydrocortisone	MuS severity	Not clear for this condition	[108]
	Serum	UHPLC-MS	Tryptophan	MuS severity	Not clear for this condition	[108]
	Serum	UHPLC-MS	Glutamate	MuS severity	Related to excitatory neurotransmitter and oligodendrocyte death in the white matter	[108]
	Serum	UHPLC-MS	Eicosapentaenoic acid	MuS severity	Related to the activation of the immune system	[108]
	Serum	UHPLC-MS	13S-hydroxyoctadecadienoic acid	MuS severity	Not clear for this condition	[108]
	Serum	UHPLC-MS	Lysophosphatidyl cholines	MuS severity	Present in the cell membrane; role in proliferative growth and apoptosis	[108]
	Serum	UHPLC-MS	Lysophosphatidyl ethanolamines	MuS severity	Not clear for this condition	[108]
	Serum	GC-MS	Laurate	Differentiate MuS from healthy subjects	Saturated fatty acid, may be related to immune response	[109]
	Serum	GC-MS	N-methylmaleimide	Differentiate MuS from healthy subjects	May be related to mitochondrial function and energy metabolism	[109]
	Serum	GC-MS	Acylcarnitine C14:1	Differentiate MuS from healthy subjects	Related to mitochondrial function and energy metabolism	[109]
	Serum	GC-MS	Phosphatidylcholine	Differentiate MuS from healthy subjects	Present in cell membrane and myelin	[109]
PD	CSF; Urine; Brain of goldfish homogenate	GC-MS/LC-MS; NMR-based	BCAA	Differentiate PD from healthy subjects; Idiopathic PD prediction; PD Goldfish model	Protein synthesis, energy production, and synthesis of the neurotransmitter glutamate	[6,143,145]
	Serum	UPLC-MS/MS	Caffeine	Differentiate PD from healthy subjects	Regulate the release of neurotransmitters (glutamate and dopamine)	[133]
	Serum	UPLC-MS/MS	Tryptophan	Differentiate PD from healthy subjects	The decrease may be associated with psychiatric problems in advanced PD	[133]
	Serum	UPLC-MS/MS	Ergothioneine	Differentiate PD from healthy subjects	A decrease may suggest elevated oxidative stress	[133]
	Serum	UPLC-MS/MS	Bilirubin/Biliverdin ratio	Differentiate PD from healthy subjects	A decrease may suggest elevated oxidative stress	[133]
	Serum; Plasma	Enzymatic Methods	Uric acid	PD prediction	Antioxidant. An increase may suggest a potential protective effect	[136,137,138]
	Serum	MS-based	FA metabolism (acyl carnitine pathway)	PD prognosis and MCI development	Medium-long chain FA derived from beta-oxidation. Related to mitochondrial dysfunction and neuronal loss	[142]
	Urine	HPLC-HRMS	Steroidogenesis metabolism	PD progression	May be related to oxidative stress, inflammation, and neuron injury	[143]
	Urine	HPLC-HRMS	Fatty acid beta-oxidation	PD progression	May be related to mitochondrial dysfunction, oxidative stress, and impaired energy metabolism	[143]
	Urine	HPLC-HRMS	Histidine metabolism	PD progression	Suppressive neurotransmitter effects, and hormone secretion	[143]
	Urine	HPLC-HRMS	Phenylalanine metabolism	PD progression	Not clear for this condition	[143]
	Urine	HPLC-HRMS; GC-MS /LC-MS	Tryptophan metabolism	PD progression; Idiopathic PD prediction	Related to mitochondrial disturbances and impairment of brain energy metabolism	[143,144]
	Urine	HPLC-HRMS; GC-MS/ LC-MS	Glycine derivation	PD progression; Idiopathic PD prediction	Stimulate the release of dopamine and acetylcholine	[143,144]
	Urine	HPLC-HRMS	Nucleotide metabolism	PD progression	Not clear for this condition	[143]
	Urine	HPLC-HRMS	Tyrosine metabolism	PD progression	Not clear for this condition	[143]
	Urine	GC-MS/ LC-MS	Steroid hormone biosynthesis	Idiopathic PD prediction	Related to oxidative stress, and dopamine cell degeneration in PD	[144]
	Urine	GC-MS/ LC-MS	Phenylalanine metabolism	Idiopathic PD prediction	Precursor for dopamine	[144]
	Brain of goldfish homogenate	NMR-based	Myo-inositol	PD Goldfish model	Glial marker. An increase may suggest disruptive cell functions in the brain	[145]
	Brain of goldfish homogenate	NMR-based	N-acetylaspartate	PD Goldfish model	The decrease may suggest neuronal dysfunction or cell loss	[145]
	Brain of goldfish homogenate	NMR-based	Betaine	PD Goldfish model	Reduced may suggest a reduced antioxidant capacity	[145]
	Brain of goldfish homogenate	NMR-based	Phosphatidylcholines	PD Goldfish model	Component of cellular membranes. Decrease related to membrane damage	[145]
	Brain of goldfish homogenate	NMR-based	Creatine and phosphocreatine	PD Goldfish model	The decrease can be related to severe oxidative damage and energy impairment	[145]
	Brain of goldfish homogenate	NMR-based	Cholesterol	PD Goldfish model	The decrease may be related to elevated oxidative stress; impaired brain mitochondria	[145]
	Brain of goldfish homogenate	NMR-based	Polyunsaturated fatty acid	PD Goldfish model	The decrease may be associated with elevated oxidative stress	[145]
	CSF	UHPLC/ GC-MS	Benzoate	PD progression	Derived from the catabolism of phenylalanine	[139]
	Plasma	UHPLC/ GC-MS	Theobromine	PD progression	Phenylalanine metabolism	[139]
	Plasma	UHPLC/GC-MS	Theophylline	PD progression	Metabolites of the purine compound caffeine	[139]
	Plasma	UHPLC/GC-MS	Paraxanthine	PD progression	Metabolites of the purine compound caffeine	[139]
	Plasma	UHPLC/GC-MS	1-methylxanthine	PD progression	Metabolites of the purine compound caffeine	[139]
	Plasma	UHPLC/GC-MS	5-dodecanoate	PD progression	Fatty acid metabolism	[139]
	Plasma	UHPLC/GC-MS	3-hydroxydecanoate	PD progression	Fatty acid metabolism	[139]
	Plasma	UHPLC/GC-MS	Docosadienoate	PD progression	Fatty acid metabolism	[139]
	Plasma	UHPLC/GC-MS	Docosatrienoate	PD progression	Fatty acid metabolism	[139]
Stroke	Serum	GC-MS/ LC-MS	Isoleucine	Differentiate AIS from healthy subjects	Signaling molecule to regulate the growth, repair, and maintenance of the brain functions	[9]
	Serum	GC-MS/ LC-MS	Serine	Differentiate AIS from healthy subjects	Signaling molecule to regulate the growth, repair, and maintenance of the brain functions	[9]
	Serum	GC-MS/ LC-MS	Phosphatidylcholine	Differentiate AIS from healthy subjects	Component of cellular membrane	[9]
	Serum	GC-MS/LC-MS	Betaine	Differentiate AIS from healthy subjects	Part of the choline pathway; part of the antioxidant process	[9]
	Serum	GC-MS/LC-MS	Lysophosphatidylethanolamine	Differentiate AIS from healthy subjects	Component of cellular membrane	[9]
	Serum	GC-MS/LC-MS	Carnitine	Differentiate AIS from healthy subjects	Help the catabolism of lipids and energy conversion	[9]
	Serum; Plasma/Urine	GC-MS; NMR-based	Lactate	AIS prediction Small vessel disease prediction	An increase may indicate anaerobic glycolysis, hypoxia, and ischemia	[10,170]
	Serum	GC-MS	Tyrosine	AIS prediction	A low level can lead to oxidative stress and inflammation	[10]
	Serum; CSF	GC-MS	Tryptophan	AIS prediction; Long-term outcome of subarachnoid hemorrhage	A low level can reduce serotonin	[10,175]
	Plasma	HPLC	Dimethylarginine	Early-onset stroke	Inhibitor of nitric oxide synthase, part of the pathogenesis of atherosclerosis	[161]
	Plasma	NMR-based	Choline	Carotid artery stenosis pathogenesis	Its reduction increases the homocysteine methylation pathway	[162]
	Plasma	NMR-based	Homocysteine	Carotid artery stenosis pathogenesis	The increase could be associated with oxidative stress in vascular cells and platelet adhesion	[162]
	Plasma	LC-MS	Lysophosphatidylcholine	Stroke recurrence; Large artery atherosclerosis	It may be a potential trigger of the brain inflammation processes	[163,171]
	Serum	LC-MS/MS	Acetyl-L-lysine	Thrombotic ischemic prediction	The decrease may suggest elevated lysine catabolism and excitotoxic activity	[165]
	Serum	LC-MS/MS	Cadaverine	Thrombotic ischemic prediction	The decrease may suggest elevated lysine catabolism and excitotoxic activity	[165]
	Serum	LC-MS/MS	2-oxoglutarate	Thrombotic ischemic prediction	The decrease may suggest elevated lysine catabolism and excitotoxic activity	[165]
	Serum	LC-MS/MS	Nicotinamide	Thrombotic ischemic prediction	The decrease may suggest elevated lysine catabolism and excitotoxic activity	[165]
	Serum	LC-MS/MS	Valine	Thrombotic ischemic prediction	A decrease may suggest an excitotoxic activity	[165]
	Plasma; CSF	LC-MS; GC-MS	BCAA	Stroke outcome and severity; Long-term outcome of subarachnoid hemorrhage	Decreased may influence the bioenergetic homeostasis and impair the citric acid cycle pathways	[168,175]
	Plasma/ Urine	NMR-based	Pyruvate	Small vessel disease prediction	The increase may be related to anaerobic glycolysis	[170]
	Plasma/ Urine	NMR-based	Glycolate	Small vessel disease prediction	The increase may be related to folic acid deficiency and hyperhomocysteinemia	[170]
	Plasma/ Urine	NMR-based	Formate	Small vessel disease prediction	The increase may be related to folic acid deficiency and hyperhomocysteinemia	[170]
	Plasma/ Urine	NMR-based	Glutamine	Small vessel disease prediction	The decrease may be related to elevating of glial fibrillary acidic protein and brain damage	[170]
	Plasma/ Urine	NMR-based	Methanol	Small vessel disease prediction	The decrease may be related to hyperhomocysteinemia	[170]
	Plasma	HPLC	Taurine	Stroke prognosis and recovery	Osmoregulator and neuromodulator. The increase may be related to brain tissue damage	[172]
	Blood	Mobile Photometric - Enzyme-kinetic Analyzer	Lactate:Pyruvate ratio	Hemorrhagic stroke prognosis	Reduced pyruvate may be related to impairment in energetic and repair functions	[174]
	CSF	GC-MS	2-hydroxyglutarate	Long-term outcome of subarachnoid hemorrhage	The increase was related to adverse outcome and death, while the decrease was related to low disability outcomes	[175]
	CSF	GC-MS	Glycine	Long-term outcome of subarachnoid hemorrhage	Not clear for this condition	[175]
	CSF	GC-MS	Proline	Long-term outcome of subarachnoid hemorrhage	Not clear for this condition	[175]

CSF—Cerebrospinal Fluid; MS—Mass Spectrometry; LC/MS—Liquid Chromatography Mass Spectrometry; GC/MS—Gas Chromatography Mass Spectrometry; FIA/MS/MS—Flow Injection Analysis using tandem Mass Spectrometry; MCI—Mild Cognitive Impairment; UPLC MS/MS—Ultra Performance Liquid Chromatography—Tandem Mass Spectrometer; NMR—Nuclear magnetic Resonance; SOD1—Superoxide dismutase 1; HMRS—High-Resolution Mass Spectrometry; TCA—Tricarboxylic Acid Cycle; FA—Fatty acids; HPLC—High Performance Liquid Chromatography; AIS—Acute Ischemic Stroke.

### 2.7. Overview of Relevant Metabolites in Neurological Disorders

The first interesting observation is that there are altered metabolic processes in common among the different neurological disorders. In this context, we can identify metabolites such as glutamate and glutamine related to hyperexcitation and neuronal excitotoxicity processes—processes commonly observed in diseases such as epilepsy, ALS, and MuS, in which the phenomena of hyperexcitability and neuronal loss are present [176,177]. These metabolites are part of the glutamate–glutamine cycle, which plays a role in controlling neuronal excitability and cell viability, among other functions [81,177]. Furthermore, changes in their concentrations may also be present in stroke, where affected regions with low oxygen support produce less ATP, leading to an influx of calcium and consequent release of glutamate into the synapse [178].

Acetylcholine, which plays a central role in the functioning of the central nervous system, is another metabolite that stands-out from the metabolomic studies in neurological disorders. Several studies have demonstrated its importance by identifying changes in the concentrations of one of its biosynthetic precursors, choline. Acetylcholine has been linked to seizures in patients with epilepsy, and the cholinergic system deficiency has been identified in patients with AD and MuS [179]. Moreover, there is a correlation between cholinergic neurotransmission changes with some specific clinical manifestations, such as changes in the motor system, behavior, memory, and attention [180].

Several studies have found abnormal levels of BCAA and its metabolites in patients with PD and stroke [6,143,145,168,175], and have been linked to neurodegenerative diseases [75]. BCAA is involved in the synthesis of proteins, energy production, and the synthesis of neurotransmitters, such as glutamate [104].

Pyruvate and lactate are two metabolites involved in energy metabolism, which are frequently found in neurodegenerative disorders. Pyruvate participates in the glycolytic metabolism pathway, and it can be used in either aerobic oxidation or anaerobic glycolysis. Following the anaerobic glycolysis pathway, it is a substrate to form lactate. Following the aerobic oxidation pathway, pyruvate is a substrate to the citric acid metabolism (also known as tricarboxylic acid—TCA—cycle) [170]. It has been suggested that pyruvate and lactate play a role in the ischemic process occurring in patients with stroke, and are related to the exacerbation of the anaerobic pathway and impairment in the TCA cycle and its correlated functions and metabolites [170]. In ALS, this imbalance in the energetic metabolism was suggested as well [5]. However, lactate was found in acute MuS demyelinated plaques, which could not be explained by the anaerobic glycolysis mechanism [102]. Another metabolite related to energy impairment is acetone. This ketosis end-product was found in studies of MuS and ALS and may indicate impairment of energy metabolism associated with the use of acetoacetyl-CoA as a source of energy [104].

Many studies have already demonstrated the importance of some phospholipids such as lysophosphatidylethanolamine and sphingomyelin to assess the degree of neuronal damage and changes in the myelin sheath potential predictors of the stage of diseases such as AD, PD, and MuS [70,109,111]. In addition, it is also possible to observe the involvement of harmful oxidative processes, with the potential to affect membranes, as has already been proposed for Lysophosphatidic acid C18: 2 in patients with AD [71]. Furthermore, carnitine and the related metabolite acetylcarnitine, an intermediary of fatty acid oxidation, were found in studies on AD, ALS, MS, and stroke and related to impairment mitochondrial and energy metabolisms and fatty acids transport during the catabolism of lipids [9,68,83,109]. Because of the acetyl group, the acetylcarnitine can cross the blood–brain barrier more easily and is the preferred form in the central nervous system [9]. Moreover, carnitine and acetylcarnitine have neurotrophic, neuroprotective, and neuromodulator properties that may be important in neurodegenerative disorders [9].

Interestingly, it is possible to detect several similarities in the metabolic profile of different neurodegenerative disorders [9]. These are mainly related to energetic metabolism, oxidative pathways, and cellular membrane [9]. Although the above-mentioned findings may indicate a common mechanism present in different conditions in which neurodegeneration is present, it may also indicate a limitation of metabolic studies to discriminate the various biological pathways involved in neurodegeneration.

## 3. Perspectives and Conclusions

Neurological diseases are complex and multifactorial; they significantly impact the patient’s quality of life. Therefore, discovering new approaches that can assist in the early diagnosis, establishing prognosis, and monitoring treatment response is of paramount importance. Metabolites can be used as disease biomarkers, and metabolomic studies have the potential to discover candidate molecules that could be used as noninvasive biomarkers of neurological disorders. As they can be found in circulation, they are easily accessible with minimally invasive procedures. This factor represents a significant additional advantage for the clinical setting application, especially if one considers the limitations on accessing disease tissue in most neurological disorders.

In this review, we have described the main data regarding metabolomics studies on neurological diseases. Despite the advanced research in this field, a large volume of the information generated came from exploratory studies and the reporting of new methods rather than work on discovery and validation of biomarkers for clinical use [26]. Indeed, most studies rely on exploratory approaches to better understand disease mechanisms [26]. Even though high-throughput metabolomics has not identified a definite metabolic biomarker to assist in the clinical management of patients with neurological disorders, these broad approaches may generate a list of candidate molecules and characteristic metabolic profiles for the disease, which should be subsequently used for validation studies before going into clinical application.

Furthermore, the metabolomic profile is influenced by several environmental and biological factors, which may be unrelated to the cause of the disease, leading to confounding factors in interpreting the results [154]. Certainly, unspecific individual variability is one of the biggest challenges in applied medical research, and all efforts should be made to decrease its impact on the final results. Unspecific variability or confounding factors not related to the cause of the disease under investigation can be introduced in all steps of a metabolomic study, from cohort selection to sample preparation and data analysis. As a result of this, reliable early study design and qualified data interpretation are essential to avoid critical pitfalls.

It also should consider the current lack of information on the function of candidate biomarkers identified, requiring additional in vitro and in vivo metabolomic studies [154]. The development of new and advanced technologies to identify unknown metabolites can help to solve this issue. More sensitive and accurate methods allow the identification of less concentrated molecules, which could still lead to effects on disease mechanism. It is also important to highlight recent metabolomic studies performed in tissue from genetically manipulated animal models of disease [33,34,35,36]. These may not only help to elucidate disease mechanisms but can also identify targets for genetic therapies. In the in vitro area, the relevance of single-cell studies has increased in the past few years due to their potential to interpret each cell type’s metabolic profile separately. Nevertheless, as we discussed previously, this analysis needs highly sensitive and accurate methods.

Even though the technological developments in metabolomics have been impressive over the past few years, there are still important caveats to be considered. A possible source confounding evidence is the distinction between identifying a putative cause of the disease and a simple correlation between the presence of a metabolite and the disease [15]. This distinction cannot be easily solved based solely on metabolomics experiments, and usually requires additional biological studies. An additional concern with metabolomic studies is considering if the metabolites identified are directly associated with the biological event of interest or secondary to differences in genetic background, epigenetic regulation, or disturbed microbiome [15]. Indeed, the microbiota’s metabolism has emerged as a point of interest due to their ability to mediate or modify disease risk, including neurological disorders, by affecting host immunity, endocrine function, drug metabolism, and neurological signaling [181,182,183]. Understanding the microbiota’s metabolic profile and how it affects the neurological disease’s development, outcome, and treatment is still lacking, and additional studies are needed.

The complexity of the biological mechanisms on which the neurological diseases rely upon makes it very difficult to find a unique metabolite or biomarker that could distinguish a specific disease-state or predict treatment response [7,13,14]. Therefore, it is probably more productive to search for a profile of metabolites, which may better predict the phenotype and mechanisms of interest [7].

In addition, a combination of metabolomics and other types of ‘omics’ data should also be explored when searching for insights into disease mechanisms and biomarkers of disease [155]. Although very relevant for the investigation of disease mechanism, there is still a lack of multi-omics implementation, mainly due to the complexity of data integration, validation, and interpretation [184,185]. The multi-omics approach is an exciting strategy to understand the simultaneous biological mechanisms occurring during a disease. It considers an integration among imaging, genomic and regulatory variants, proteins, lipids, and metabolites, leading to a complete characterization of the disease’s mechanisms. An important limitation of the multi-omics approach applied to human samples is the presence of inter-individual variability, which may affect the ability to identify disease-specific variants. Thus, to tease apart what is a normal variation and what is genuinely associated with disease states, one needs powerful and advanced computational and statistical tools, some of which are just being developed [185]. In biologically complex diseases, such as neurological disorders, one expects an array of mechanisms, such as neuroinflammation, oxidative stress, and neurodegeneration, to name a few, and there will be the need to establish some hierarchical classification among them to understand better the flux of events leading to disease and, most notably to identify potential therapeutic targets. Despite all these issues, the multi-omics strategy is a promising approach to enable additional clinical decision-making accuracy on treatment and diagnosis [13]. Thus, metabolomics can become an essential tool in precision medicine.

In summary, recent studies have identified metabolites as potentially associated with neurological diseases. The development of new technologies allows for the analysis of smaller and less concentrated molecules with high efficacy, enabling the development of studies on single-cell metabolomics, which, together with the recent investigation of the metabolomics of the microbiota and multi-omics analysis, will likely provide a never seen broader view of the influences of metabolites in disease. However, additional studies are required in metabolomics, especially those aimed at translating basic research into clinical practice and the implementation of metabolomics techniques in the clinic [156].

## Figures and Tables

**Figure 1 metabolites-10-00389-f001:**
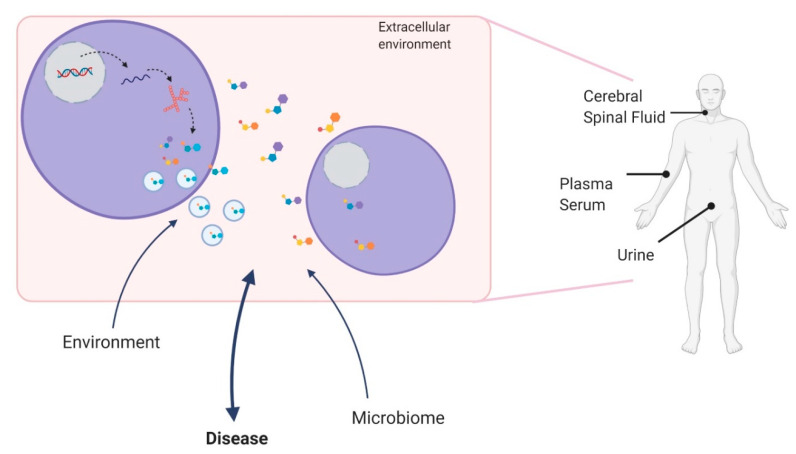
**Circulating metabolites**. Metabolites are small molecules that act locally, in the tissue where they are produced, or systemically, when released to the extracellular environment. In the latter case, metabolites can circulate and be identified in different biofluids, such as plasma serum, urine, and cerebral spinal fluid. Circulating metabolites can be released as free molecules or inside vesicles. The production of metabolites can be influenced by changes in the external or internal environment, such as the microbiome, as well as by the occurrence of disease. In this last case, the disease processes may impact metabolite production and, thus, make it possible to use metabolites as disease biomarkers. The figure was created with Biorender.com.

**Figure 2 metabolites-10-00389-f002:**
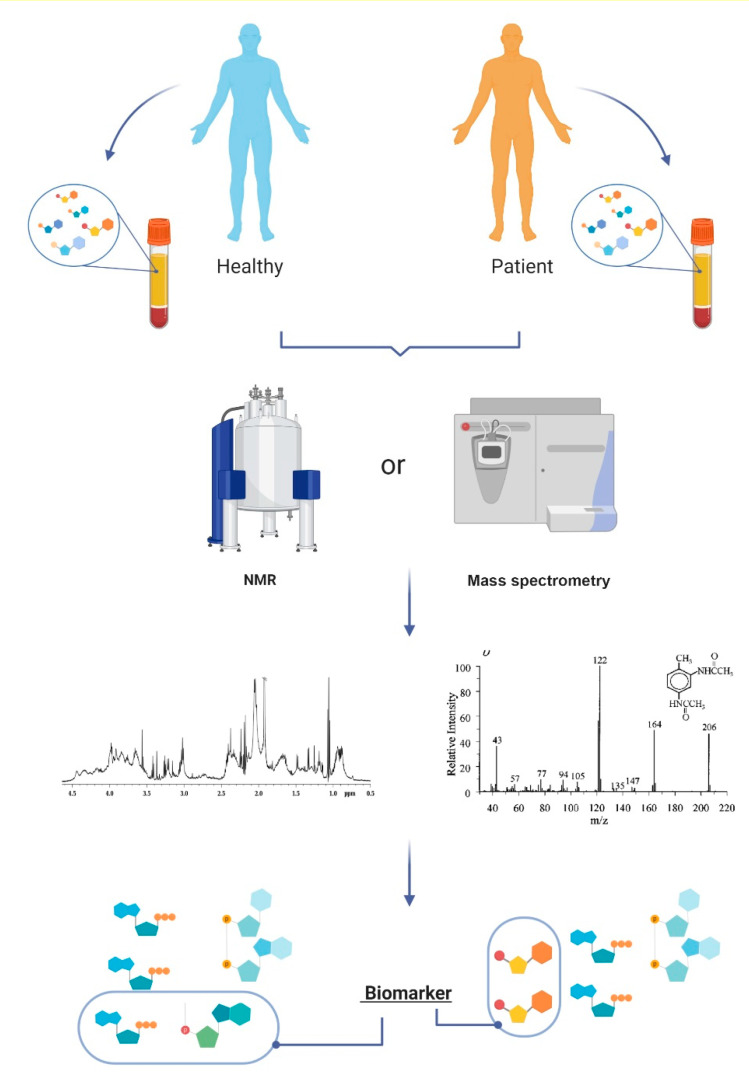
**Metabolomic scheme for identifying biomarkers**. Circulating metabolites are found in biofluids and may be used as disease biomarkers. Metabolomics can be applied to identify metabolites using high-throughput techniques, such as mass spectrometry or nuclear magnetic resonance (NMR). By processing and analyzing the data from metabolomics studies, one can identify specific metabolites as disease biomarkers. This figure was created with Biorender.com.

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
