# Peer review of "Circulating Metabolites as Potential Biomarkers for Neurological Disorders—Metabolites in Neurological Disorders"

_metabolites, 2020, doi:10.3390/metabo10100389_

Round 1

Reviewer 1 Report

The paper by Donatti and colleagues gives an overview of the knowledge related to the potential of circulating metabolites as biomarkers in the diagnosis of neurological disorders. The paper is well written and I have only few minor comments or advice for ameliorating the paper.

General remarks:

I would suggest that a native English speaker or language professional reviews the manuscript prior to resubmitting the article.

it would be useful for the authors to briefly present and summarize in the table the metabolites they referred to in the text as potential biomarkers of certain neurological disorders

Minor remarks:

Please rephrase the sentences to make them more understandable:

Line 107-108: “In contrast, LC has greater chemical versatility, allowing a multivariate analysis of specific analytes, so as not to require sample heating or derivatization reactions [4].”

Line 232-233: “Hence, one study utilized H1NMR spectroscopy to examine the metabolite profile of serum from patients with ALS there was increased glutamate [29].”

Please correct the following:

Line 104: “…coupling WITH a mass spectrometer”

Line 181-182: “Besides, AD and MCI patients showed differences in lysophospholipid concentrations, WITH higher levels in AD patients and lower levels in MCI patients [21].”

Line 231: “…glutamate as an important circulating metabolite IN patients with ALS [24,27,29,30].”

Line 259-260: “One study performed a targetED metabolomic ANALYSIS by tandem mass spectroscopy (MS/MS)…”

Line 265-266: “Glutamic acid and glycine levels were correlated to disease progression IN THE GROUP RECEIVING olesoxime treatment…”

Line 294-296: Please cite the reference here.

Line 339-340: “It affects the CNS, and although it IS typically presentED as episodes of clinical deterioration…”

Line 370: “An interesting recent work proposed a new method to be applied FOR STUDYING lipidomics…”

Line 409: “…of A2A adenosine A2A receptors…”, please delete the first A2A.

Line 486: “…increasing IN young individuals [92].”

Line 527-528: “Each subtype has different etiologies, and thus specific metabolites may BE present in different TOAST sub-types.”

Line 575: “…metabolic biomarkers will beCOME an important tool…”

Line 591-592: Conflict of interest - the same statement appears twice.

Please give the full name when you first mention the abbreviation:

Line 218: SOD1 gene (gene coding for superoxide dismutase 1)

Line 401: PINK1 gene (gene coding for PTEN-induced kinase 1)

Please elaborate more the statement in line 151-152 regarding the diagnosis of AD. Even though beta-amyloid deposits in the brain are a characteristic feature of AD, their presence cannot be used to diagnose the disease. The definitive conformation of AD can be achieved only after death, by linking clinical measures with an examination of brain tissue in an autopsy (measuring the frequency of neuritic plaques and neurofibrillary tangles in different brain regions).

Author Response

We appreciate all your suggestions. Please, see the attachment for all replies. 

Reviewer 2 Report

The review was mainly written using a descriptive method. I find that it is hard to be accepted for a publication in the current form. Therefore, I have several comments to hopefully improve the quality of the review:

  1. A review should not simply describe what has been done without any clear discussion on which progress has been achieved. All the main results of the cited documents could be simply presented in the form of multiple tables.
  2. It would be much better if some conceptual suggestions can be made so that subsequent investigations can avoid current pitfalls. There are several excellent review papers on this regard by Dr. Warwick Dunn, Dr. Oliver Fiehn, and Dr. Sung Won Kwon among others.
  3. The technical difficulties of metabolomics are not presented in the current form of the review, it may be better to be included.
  4. In the perspective and conclusions section, it would be better if concrete suggestions could be provided for subsequent studies to facilitate the utilization of metabolomics in clinics.
  5. I would like to see some discussion on frequently reported metabolites in papers that are covered by this review. It will be helpful for other authors to possibly develop targeted assays.
  6. Please kindly improve the readability of the manuscript and check all typos, e.g., line 591-592.
  7. This review does not look comprehensive to my eyes, I suggest more reference search to improve the coverage of the review.

Author Response

We would like to thank you for all the comments and suggestions. Please, see the attachment for all replies. 

Round 2

Reviewer 2 Report

The manuscript is significantly improved even though the discussion toward the advances of metabolomics is still light.

As in my previous recommendations, more technical discussion on the metabolomics from the leading scientists of the field will be helpful.

Also, unique challenges in applying metabolomics and lipidomics of this group of diseases are better to be mentioned: (1l Pitfalls in using the metabolic profiles to elucidate the mechanistic information of the disease. (2) The lack of multi-omics implementation. (3) Problematic quantification.

Round 3

Reviewer 2 Report

The manuscript has significantly improved.